

# Simulated Long-term Evolution of the Thermosphere during the Holocene: 1. Neutral Density and Temperature

Yihui Cai[1,2,3], Xinan Yue[1,2,3], Xu Zhou[1,3], Zhipeng Ren[1,2,3], Yong Wei[1,2,3], Yongxin Pan[1,2,3]

[1]Key Laboratory of Earth and Planetary Physics, Institute of Geology and Geophysics, Chinese Academy of Sciences, Beijing, 100029, China
[2]College of Earth and Planetary Sciences, University of Chinese Academy of Sciences, Beijing, 100029, China
[3]Beijing National Observatory of Space Environment, Institute of Geology and Geophysics, Chinese Academy of Sciences, Beijing, 100029, China

*Correspondence to*: Xinan Yue (yuexinan@mail.iggcas.ac.cn)

**Abstract.** In the previous work of Yue et al. (2022), the ionospheric evolution during the Holocene (9455 BC to 2015 AD) was comprehensively and carefully investigated for the first time using the Global Coupled Ionosphere-Thermosphere-Electrodynamics Model developed at the Institute of Geology and Geophysics, Chinese Academy of Sciences (GCITEM-IGGCAS), driven by realistic geomagnetic fields, $CO_2$ levels, and solar activity derived from the ancient media records and modern measurements. In this study, we further quantify the effects of the three drivers on thermospheric neutral density and temperature variations during the Holocene. We find that the oscillations of solar activity contribute more than 80% of the thermospheric variability, while either $CO_2$ or the geomagnetic field contributes less than 10%. The effect of $CO_2$ on the global mean neutral density and temperature is comparable to that of the geomagnetic field throughout the Holocene but is more significant after 1800 AD. In addition, thermospheric density and temperature show approximately linear variations with the dipole moment of the geomagnetic field, $CO_2$, and F10.7, with only the linear growth rate associated with the geomagnetic field varying significantly in universal time and latitude. The increasing dipole moment and $CO_2$ cool and contract the thermosphere, while solar activity has the opposite effect. The higher the altitude, the greater the influence of the three factors on the thermosphere. Different factors produce different seasonal variations in thermosphere changes. Furthermore, we predict that a 400 ppm increase in $CO_2$ will result in a 50–70% and 84–114 K reduction in global mean neutral density and temperature, respectively, which should directly affect the orbit and lifetime of spacecraft and space debris.

## 1 Introduction

Global glaciers have been melting in the recent century due to climate warming, and this melting has been accelerating in the last 20 years, leading to rising sea levels and elevating natural disasters (Hugonnet et al., 2021; Zemp et al., 2019). The main cause of climate warming is the use of fossil fuels as a source of anthropogenic greenhouse gases (Tollefson, 2021). The simulations of Roble and Dickinson (1989) show that the increase in greenhouse gases warms the troposphere, but cools the thermosphere. The long-term trend studies in subsequent decades largely support this consensus (Laštovička, 2009; Laštovička et al., 2006; Laštovička et al., 2008). Tropospheric warming seriously affects human life, while changes in the thermosphere



affect various human-launched satellites, space stations, and spacecraft, such as the SpaceX Starlink satellites destruction event on 4 February 2022 (Dang et al., 2022; Lin et al., 2022), therefore is also relevant to human life. The current understanding of the thermosphere is based on modern satellite observations over the last ~70 years. The energy sources driving the variability of the thermosphere include mainly solar irradiance and geomagnetic activity generated by the interaction between the solar wind and the Earth's magnetic field (Knipp et al., 2004). These two energy sources are responsible for thermospheric temporal variations on time scales ranging from minutes to decades. On longer time scales, however, the effects of greenhouse gases and the geomagnetic field must be taken into account (Laštovička et al., 2006). Several review papers have summarized the knowledge of thermospheric variations and their driving mechanisms (Laštovička, 2017; Laštovička et al., 2012; Qian et al., 2011; Qian & Solomon, 2012).

Long-term trends of neutral density at different altitudes between 200 and 600 km have been extensively investigated using satellite orbit measurements since the 1960s (Emmert, 2015; Emmert et al., 2004; Emmert et al., 2008; Keating et al., 2000; Marcos et al., 2005; Saunders et al., 2011). These studies suggested that the trend is mainly attributed to a dramatic increase in anthropogenic greenhouse gases and becomes stronger with increasing altitude. A summary of the trends derived from satellite orbit data can be found in Emmert (2015) and Solomon et al. (2018). Overall, the observed long-term trend of the thermospheric neutral density at 400 km ranges from –2% to –5% per decade. These observed characteristics are qualitatively consistent with the model-predicted effects of increasing $CO_2$ concentrations. (Qian et al., 2006; Roble & Dickinson, 1989; Solomon et al., 2015). The effect of geomagnetic field strength and configuration on the thermosphere has also been paid enormous attention from observations and simulations (A et al., 2012; Cnossen, 2014; Cnossen, 2022; Cnossen & Maute, 2020; Cnossen et al., 2012; Cnossen et al., 2011; Förster & Cnossen, 2013). However, none of these effects are as significant as the effect of solar activity on the thermosphere. The amplitude of the solar-driven variation increases with height, by a factor of two for temperature and an order of magnitude for density in the upper thermosphere (Qian & Solomon, 2012; Solomon et al., 2019). Overall, these observations and simulations of the thermosphere are limited to the recent 100 years, while the "ancient" thermosphere has never been investigated on longer time scales such as during the Holocene. Therefore, we propose to reconstruct the paleo-thermosphere since the Holocene using the first principle numerical model driven by these indices including solar activity derived from tree ring (Solanki et al., 2004), paleo geomagnetic field model (Korte et al., 2011), and greenhouse gas concentration derived from polar ice core (Lüthi et al., 2008). Based on the reconstructed simulations, we can understand in detail the evolution of the thermosphere over 10,000 years and the effects of changes in the geomagnetic field, $CO_2$, and solar activity on the thermosphere, which can provide a foundation for the future effect of climate change on human life.

The rest of the paper is organized as follows. Section 2 will briefly describe the numerical model and driving parameters used in this study. Section 3 will show the simulation results and discuss them. Finally, we draw conclusions in Section 4.



## 2 Methodology

This study will use the Global Coupled Ionosphere-Thermosphere-Electrodynamics Model developed at the Institute of
Geology and Geophysics, Chinese Academy of Sciences (GCITEM-IGGCAS) (Ren et al., 2009), which is the same as Yue et
al. (2022). This model self-consistently solves the energy, momentum, and continuity equations of neutrals and ions in altitude
coordinate rather than pressure level coordinate between 90 and 600 km, and solves the electrodynamic equations using
magnetic apex coordinate (Richmond, 1995) based on provided spherical harmonic coefficients of any dipole-dominated
geomagnetic field, such as the International Geomagnetic Reference Field (IGRF) model (Alken et al., 2021). This model has
a good performance that has been confirmed by several ionospheric and thermospheric weather and climate simulations (Ren
et al., 2011; Ren et al., 2020; Ren et al., 2010; Yue et al., 2022; Zhou et al., 2022). To remove the effects of initial conditions,
each simulation is run for an interval of 15 days, and the final day results are used for further analysis.

Three drivers associated with geomagnetic field, $CO_2$ level, and solar activity during the Holocene, 9455 BC to 2015 AD, are
used to drive the GCITEM-IGGCAS. These drivers have been summarized in Figure 1 of Yue et al. (2022). The geomagnetic
field is CALS10k.2 (Constable et al., 2016) for the period 9455 BC to 1900 AD and IGRF after 1990. The geomagnetic field
has undergone complex and nonlinear changes during the Holocene, with a dipole moment variation of ~40%, much larger
than the ~7% variation since 1900. The $CO_2$ concentration evolution that is derived from Antarctica Vostok and EPICA Dome
C ice cores (Lüthi et al., 2008), Antarctica Law Dome ice cores (Macfarling Meure et al., 2006), and direct atmospheric
measurement at Mauna Loa Observatory, Hawaii (Keeling et al., 1995). The $CO_2$ concentration increases roughly linearly
from 250 ppm (parts per million) around 10,000 BC to 402 ppm around 2015 AD, with a major increase occurring after 1800.
The F10.7 index evolution converted from the tree ring derived sunspot number (SSN) (Solanki et al., 2004) and the group
SSN (Hoyt & Schatten, 1998), and modern instrument measurement (Tapping, 2013), which reveals the long-term oscillations
in solar activity with relatively high solar activity around 9000 BC and 1960 AD. A detailed description of the three drivers
can be found in Yue et al. (2022).

In this simulation, four control runs (CR1–CR4) have been implemented, which is the same as Yue et al. (2022) (summarized
in their Table 1). CR1 is used to identify the effect of geomagnetic field variation on thermospheric evolution. CR2 is used to
reveal the effect of $CO_2$. CR3 is used to diagnose the combined effect of geomagnetic field and $CO_2$. CR4 is used to determine
the combined effects of geomagnetic field variation, solar activity, and $CO_2$. In addition, the combined analysis of the
simulations of CR3 and CR4 allows for discerning the effect of solar activity.

## 3 Results and Discussions

### 3.1 Thermosphere evolution during Holocene

Figure 1 shows the overall evolution of the CR4 simulated thermospheric neutral density and temperature in the March equinox
during the Holocene. The top row displays the time evolution of the global mean neutral density and temperature at 400 km,



and the other rows give the global map of neutral density and temperature at 400 km at Universal Time (UT) 19:00 for the
considered four years (9005 BC, 7635 BC, 3015 BC, and 2005 AD). The white text of the global map in the left column of
Figure 1 gives the corresponding F10.7 index, $CO_2$ level, and dipole moment of the geomagnetic field. It is clear that the global
mean neutral density and temperature are mainly controlled by solar activity as they show essentially the same temporal
evolution and oscillation as solar activity. The higher the solar activity, the larger the neutral density and temperature in general,
with a relatively larger value around 9000 BC and 1960 AD. The global distribution of neutral density and temperature shows
a remarkable feature during the dayside with two peaks on either side of the geomagnetic inclination equator, so-called
equatorial mass anomaly (EMA), like the equatorial ionization anomaly (EIA) in the ionosphere (Appleton, 1946; Balan et al.,
2018), indicating a strong coupling between the thermosphere and ionosphere modulated by the geomagnetic field at low and
middle latitudes (Hedin & Mayr, 1973; H. Liu et al., 2005; Huixin Liu et al., 2007; Raghavarao et al., 1993; Raghavarao et al.,
1991). Comparing the simulations for the considered four years can basically reveal the influence of $CO_2$, geomagnetic field,
and solar activity on the evolution of the thermosphere. The F10.7 index and $CO_2$ level used to drive GCITEM-IGGCAS are
similar in 9005 BC and 7635 BC, except that the dipole moment is about 20% larger in 7635 BC than in 9005 BC. It is clear
that the simulated neutral density and temperature for these two years are not significantly different in global distribution
pattern and magnitude, which indicates that the ~20% change in dipole moment has a weak effect on the thermosphere at 400
km altitude. Comparing the simulations of 9005 BC and 3015 BC reveals that a ~25% reduction of the F10.7 index leads to a
~50% decrease in neutral density and a ~170 K decrease in neutral temperature. Furthermore, the cooling effect of $CO_2$ on the
thermosphere can be found by comparing the simulations of 9005 BC and 2005 AD. An increase of 110 ppm (~40%) of $CO_2$
concentration causes a temperature reduction of ~40 K and a density reduction of ~21%. In addition, we also checked the
simulations at other UTs and during the June solstice as well. In summary, the thermosphere is primarily controlled by solar
activity, with secondary controlled factors being $CO_2$ and geomagnetic field.

Figure 2 shows the global mean for all UTs and grids of the neutral density and temperature as a function of altitude and year
(left column) and the zonal mean neutral density and temperature versus latitude and year at 400 km (right column) in March
equinox during the Holocene. According to (Afraimovich et al., 2008), a latitude-dependent area weighting factor was used in
the calculation of the global mean to make it more representative. As shown in Figure 2, both the global mean and zonal mean
display significant oscillations throughout the whole Holocene, which is consistent with the oscillations of the solar activity
whose relative change was marked by the red lines in the left column. When F10.7 reaches its relatively higher value before
8000 BC and in the recent century, a larger value of neutral density and temperature also appear. This feature is not clear in
the altitude profile of the global mean of neutral density (top left panel) due to the span of about 10 orders of magnitude.
However, this does not affect the conclusion, after all, the fixed height of the zonal mean is more revealing of this feature.
Furthermore, the thermospheric neutral density and temperature also show significant long-term decreases from 6000 BC to
3500 BC and the most famous grand solar minimum (Usoskin et al., 2007), the Maunder Minimum between 1645 and 1715
(Eddy, 1976). Only a weak latitudinal variation can be seen in the zonal mean neutral temperature in the March equinox.



## 3.2 The effects of the three drivers on the evolution of thermospheric neutral density and temperature

In this section, the changes in thermospheric neutral density and temperature caused by the geomagnetic field, $CO_2$, and solar activity will be diagnosed by subtracting the beginning year (9455 BC) of the simulations, as shown in Figures 3–5 for the results of the global mean and zonal mean.

Figure 3 shows the global mean neutral density profile variations in percentage caused by the three drivers in the left column. The black, magenta, and red lines represent the relative changes of the diploe moment, $CO_2$, and F10.7 index, respectively, during the Holocene. In general, the higher altitude, the larger the effect of the three drivers on the neutral density, because the neutral density decreases exponentially with altitude causing all effects to be amplified at higher altitudes. For the effect of the geomagnetic field, its nonlinear variation causes a nonlinear change in the neutral density, and a decrease in its intensity represented by the weakening of the dipole moment (around 5500 BC) generally leads to an increase in the neutral density. In addition, an increase in the dipole moment would make the neutral density increase weaker, which might be related to the decrease in Joule heating due to the strong dipole moment (Cnossen et al., 2012; Cnossen et al., 2011; Glassmeier et al., 2004; Wang et al., 2017). When the dipole moment increases beyond $\sim 3 \times 10^{22}$ Am$^2$ (in $\sim$7500 BC and from $\sim$1500 BC to $\sim$1000 AD), the density will decrease in turn between 150 and 250 km. For the effect of $CO_2$, the neutral density decreases during the increase phase of $CO_2$ level (before $\sim$8000 BC and after $\sim$4000 BC). This is because greenhouse gases can cool and contract the thermosphere (Qian et al., 2011), as shown in Figure 4 for temperature reduction. In turn, when $CO_2$ decreases between $\sim$8000 BC and $\sim$4000 BC, the thermosphere neutral density increases. It is worth noting that the effect of $CO_2$ has been more significant since 1800 AD due to the much larger growth rate of $CO_2$. For the effect of solar activity, the overall change in neutral density due to solar activity is more than 10 times larger than that of $CO_2$ and the geomagnetic field, so it is the dominant factor in neutral density change. The neutral density has increased by more than 100% in the recent century and around 9000 BC, which corresponds to relatively greater solar activity. The right column of Figure 3 is the zonal mean results at 400 km. The grey lines in panel (d) mark the latitude of the north and south magnetic poles for the corresponding year. The effects of the three drivers on the temporal evolution of the neutral density at all latitudes are similar to those characterized in the left column, with no significant latitude variations in the effects of $CO_2$ and solar activity, and a weak latitude variation in the effect of the geomagnetic field, which is stronger at high latitudes. The region with greater geomagnetic field effects corresponds exactly to the magnetic pole locations, such as the south magnetic pole at $\sim$64° around 5500 BC, implying the importance of the magnetic pole locations and further supporting the contribution of Joule heating in the polar region.

Figure 4 shows a similar pattern as Figure 3 except that the neutral temperature variations are shown. Overall, the effects of $CO_2$ and solar activity on temperature are essentially the same as those on neutral density, only the temperature change is more significant. The dramatic increase in $CO_2$ level in the past century has led to a decrease in global mean neutral temperature of more than 20 K, which is well in line with previous understanding (Cnossen, 2014; Roble & Dickinson, 1989). In addition, the global mean neutral temperature increases within 5 K due to $CO_2$ reduction between 8000 and 4000 BC. Solar activity remains the dominant factor in the neutral temperature variability, which leads to neutral temperature changes in the range of





±200 K with its own oscillations. Furthermore, the effect of the geomagnetic field on neutral temperature differs significantly from the effect on neutral density, but can still be explained by the geomagnetic field structure, dipole moment strength, and Joule heating. The zonal mean temperature changes contributed by the geomagnetic field show a clear latitude variation. The neutral temperature increases ~24 K at south latitude ~65° around 5500 BC, caused by a reduction in the dipole moment resulting in stronger Joule heating around the south magnetic pole. Conversely, between 1500 BC and 1000 AD, the neutral

temperature in the polar regions dropped by up to 22 K due to the weakening of Joule heating caused by the increase of the dipole moment. In addition, the effect of the geomagnetic field is significantly weaker at north latitude ~10°, perhaps owing to tides in the lower atmosphere. Although the neutral temperature changes in the polar regions are large, the change in global mean neutral temperature due to the geomagnetic field is essentially within ±10 K at all altitudes during Holocene. The June simulations have also been carefully analyzed (not shown here), and the effects of $CO_2$ and solar activity are similar to those

of March, and the geomagnetic field effects differ greatly from those of March, which lead to a weakening of both the neutral density and temperature except for an increase near 5500 BC. The contributions of geomagnetic field structure, dipole moment, and Joule heating are still evident in the June results.

Figure 5 shows the effect of the three drivers on the global mean neutral density and temperature at 400 km, characterized by the deviation obtained by subtracting the simulation results of the starting year. The grey dots represent the results for each

175 UT. Three main features can be extracted from Figure 5. (1) The oscillation range of neutral density at 400 km due to the geomagnetic field, $CO_2$, and solar activity variations is [–5 10], [–60 5], and [–200 400] $\times 10^{-14}$ kg m$^{-3}$, respectively. While it is [–10, 10], [–40, 5], and [–200, 200] K for the neutral temperature. It is clear that the effect of solar activity variations is dozens of times greater than that of $CO_2$ and geomagnetic field variations. (2) Both the neutral density and temperature decrease with increasing $CO_2$. (3) The effects of $CO_2$ and solar activity have no universal time variation, while the effects of the

180 geomagnetic field have significant universal time variation modulated by the dipole moment.

### 3.3 The long-term trends generated by the variations of the geomagnetic field, $CO_2$, and solar activity

From Figures 3–5, it can be found that the effects of the three factors vary approximately linearly. Therefore, we calculated linear growth rates for the effects of the three factors on neutral density and temperature, as shown in the text of each panel in Figure 6, which reveals the long-term trends of the thermosphere generated by the variations of the geomagnetic field (dipole

moment and the colatitude of the north magnetic pole), $CO_2$, and solar activity, respectively. The grey dots in Figure 6 are the global mean value of neutral density (left column) and temperature (right column) at 400 km in the March equinox of the corresponding year in the simulations driven by the three drivers. The red lines are the result of the least squares fitting. The neutral density and temperature show a significant linear variation with the three drivers, while the nonlinear effect of the geomagnetic field can be found in the first and second rows, shown by the scattered grey dots on both sides of the red line.

Although the fitted density is generally smaller when F10.7 is greater than 110 in the bottom left panel, the linear variation of the simulated neutral density with F10.7 is still clearly visible, only presenting a larger linear growth rate. The average value





of the global mean neutral density at 400 km over the entire simulation time interval is about $2.26 \times 10^{-12}$ kg m$^{-3}$. Based on this value and the linear growth rate shown in Figure 6, the effects due to changes in the geomagnetic dipole moment, the colatitude of north magnetic pole, $CO_2$, and solar activity during the entire Holocene can be calculated to be about –1.4%, –1.6%, –31% and +250%, respectively. While the effects are about –7 K, –4 K, –48 K, and +557 K for the neutral temperature, respectively. Solomon et al. (2019) pointed out that the temperature change from solar minimum to maximum increases by about 500 K at 400 km based on the simulation of the Whole Atmosphere Community Climate Model-eXtended (WACCM-X), which is generally consistent with our results. Since the effect of magnetic pole position is not as large as that of dipole moment, only the dipole moment is used later to quantify the effect of the magnetic field.

Figure 7 shows the altitude variations of the linear growth rate (dashed lines) and the corresponding change in percentage (solid lines) of the global mean neutral density (left column) and temperature (right column) resulting from the three drivers in the March equinox (black lines) and June solstice (red lines). The effect of the three drivers on the neutral temperatures of March and June essentially increases with altitude, but it is close to constant above 300 km. The effect of the geomagnetic field on the neutral temperature is slightly greater in June than in March. For every $10^{22}$ Am$^2$ increase in dipole moment, the neutral temperature in June decreases by ~2.7 K, compared to ~1.7 K in March. In contrast, $CO_2$ and solar activity have the opposite effect on neutral temperature, with a greater effect in March. For every 10 ppm increase in $CO_2$, the global mean neutral temperature above 200 km decreases by ~3.5 K in March and ~2.3 K in June. Akmaev and Fomichev (1998) suggested a trend of about –3.1 K per 10 ppm at 200 km in the thermosphere in April due to increasing $CO_2$, while Cnossen (2014) and Solomon et al. (2018) reported trends of about –1 K and –1.8 K per 10 ppm above 200 km, respectively. Therefore, it is reasonable that our results are 0.8–3.5 K per 10 ppm between 150 and 600 km. For each 1 sfu increase in F10.7, the neutral temperature increase is ~6.2 K and ~4.4 K in March and June, respectively. The effect of the three factors on neutral density is similar to that on neutral temperature. Since neutral density decreases exponentially with altitude, the effect of the three factors on neutral density (absolute value change) also decreases with an altitude above 150 km, as shown in the left column of Figure 7. To present more clearly the effect of the three factors on neutral density, the solid lines display the corresponding percentage changes using 2005 simulations of CR4 as a reference. This reveals that the effects of solar activity and dipole moment on neutral density increase significantly with increasing altitude in March and June with a stronger effect in June, while the effect of $CO_2$ is basically unchanged with altitude in both March and June with a stronger effect in March. In addition, the increasing dipole moment and $CO_2$ decrease the neutral density, while the rising solar activity increases the neutral density.

As mentioned in Section 3.2, only the effect of the geomagnetic field on the thermosphere displays the latitude and UT variations. Figure 8 shows the linear growth rate of neutral density (top row) and temperature (bottom row) at 400 km attributed to the geomagnetic field versus latitude and UT (left column) or longitude (right column). In general, an increase in the dipole moment attenuates the thermospheric neutral density and temperature at all latitudes, longitudes, and UTs. The linear growth rate is greater at high latitudes in the 0–8 and 18–24 UT and can reach $-3.6 \times 10^{-14}$ kg m$^{-3}$/$10^{22}$ Am$^2$ or –9.5 K/$10^{22}$ Am$^2$, while it is about $-1 \times 10^{-14}$ kg m$^{-3}$/$10^{22}$ Am$^2$ or –2 K/$10^{22}$ Am$^2$ at other latitudes and UTs. In addition, a larger linear growth rate of





up to $-4.4\times10^{-14}$ kg m$^{-3}$/10$^{22}$ Am$^2$ or $-8.7$ K/10$^{22}$ Am$^2$ is seen in all longitudes above $\pm60°$ latitude, and it is also about $-1\times10^{-14}$ kg m$^{-3}$/10$^{22}$ Am$^2$ or $-2$ K/10$^{22}$ Am$^2$ in other regions. Overall, the thermosphere in the polar regions of the Southern Hemisphere is more influenced by the geomagnetic field than that in the Northern Hemisphere, while the influence of the magnetic field is weaker and of about a similar magnitude in the middle and low latitudes.

### 3.4 Future projections

As the number of human space missions increases explosively, more and more spacecraft will operate in the thermosphere, so projecting the future state of the thermosphere is also important to human life. Based on the IPCC projections of greenhouse gas emissions under different scenarios (IPCC, 2014), we can simply and reasonably assume that $CO_2$ concentrations will rise by 400 ppm over the next century. Therefore, according to the calculations shown in Table 2, the global mean neutral density will decrease by ~70% and ~50% in March and June, respectively, due to a 400 ppm increase in $CO_2$. This is generally

consistent with the trend of about $-6.1\pm0.8$% per decade throughout the 21st century projected by Cnossen (2022). Also, from Figure 7, it can be concluded that the neutral temperatures in March and June will decrease by ~114 K and ~84 K, respectively. This is larger than the projections (~60 K) of Cnossen (2022).

The dipole moment decreases by about 3.5% over the next 50 years based on the prediction by Aubert (2015), causing an increase in global mean neutral density of up to 1% above 500 km according to the simulations of Cnossen and Maute (2020).

However, the increase of the global mean neutral density is projected to be ~0.08% and ~0.25% in March and June, respectively, based on our calculated linear growth rate. In addition, we can also project that the temperature increase due to the decrease of dipole moment in March and June is about 0.5 K and 0.7 K, respectively. As shown in Figure 8, the effect of the geomagnetic field is strongly dependent on UT and geographic location, so the global mean state projection is provided for reference only. Furthermore, the effect of geomagnetic field variation is negligible compared to the effect of rising $CO_2$ over the next 100

245    years.

### 4 Conclusions

In this study, the evolution of the thermosphere during the Holocene (from 9455 BC to 2015 AD) was simulated using the independently developed global ionosphere-thermosphere theoretical model GCITEM-IGGCAS, driven by the realistic geomagnetic field model, $CO_2$ level, and solar activity derived from modern measurements and ancient natural media.

Furthermore, through a series of control simulations, we quantify the thermospheric temperature and density changes due to variations in the geomagnetic field, $CO_2$ levels, and solar activity. The main conclusions are presented below.

1.    The climatological morphology of the global thermosphere during the Holocene is reconstructed for the first time. Thermospheric neutral density is mainly controlled by solar activity and modulated by $CO_2$ and geomagnetic field. Typically, the geomagnetic field configuration directly affects the morphology of the equatorial mass anomaly structure

of the thermosphere, while $CO_2$ mainly affects the magnitude of the neutral density and temperature. In general, the



frequent oscillations of solar activity contribute more than 80% of the thermospheric variability, while the contributions of $CO_2$ and geomagnetic field are both less than 10%. The effect of $CO_2$ is comparable to that of the geomagnetic field throughout the Holocene for the global mean neutral density and temperature but becomes more significant after 1800 AD. Only the effect of the geomagnetic field is strongly dependent on the universal time and geographical location, and

the weakening of the dipole moment leading to an increase in Joule heating in the polar region thus make the thermosphere change more than the effect of $CO_2$. Overall, the higher altitude, the larger the effect of the three drivers on the neutral density and temperature.

2.  Both the thermospheric neutral density and temperature vary approximately linearly with the dipole moment of the geomagnetic field, $CO_2$, and the F10.7 index of solar activity. The global mean variability of the neutral density at 400

265     km during the March equinox due to changes in the geomagnetic dipole moment, $CO_2$, and solar activity during the entire Holocene can be about –1.4%, –31%, and +250%, respectively. While the effects are about –7 K, –48 K, and +557 K for the neutral temperature, respectively. In addition, there is a clear altitude and seasonal variation in the thermosphere change due to an increase in per unit of dipole moment, $CO_2$, and solar activity. Different factors produce different seasonal variations in thermosphere changes. The increasing dipole moment and $CO_2$ decrease the neutral density and

temperature, while the rising solar activity increases them.

3.  We project that a 400 ppm increase in $CO_2$ will result in a 50–70% reduction in global mean thermospheric neutral density depending on the season, while neutral temperatures will decrease by 84–114 K. This is enough to change the orbit and lifetime of spacecraft and space debris, which deserves the attention of future space missions. The effect of decreasing dipole moments of the geomagnetic field over the next 100 years on the global mean thermospheric neutral density and

temperature is negligible, but the effect of changing magnetic field configurations (e.g., magnetic pole positions) on the thermosphere should be considered, especially in the polar regions.

**Data availability**

The spherical harmonic coefficients of CALS10k.2 model was obtained from the website: https://earthref.org/ERDA/2207. The IGRF model was downloaded from the website: https://www.ngdc.noaa.gov/IAGA/vmod/igrf.html. The Antarctica

Vostok and EPICA Dome C ice cores $CO_2$ level was derived from the website: https://data.noaa.gov/dataset/dataset/noaa-wds-paleoclimatology-aicc2012-800kyr-antarctic-ice-core-chronology.The Antarctica Law Dome ice core $CO_2$ data was downloaded from the website: https://www.ncei.noaa.gov/access/metadata/landing-page/bin/iso?id=noaa-icecore-9959. The Mauna Loa observed $CO_2$ was from the website: https://gml.noaa.gov/ccgg/trends/data.html. The 11,000 yr reconstructed sunspot number was downloaded from the NOAA website:

https://www.ncei.noaa.gov/pub/data/paleo/climate_forcing/solar_variability/solanki2004-ssn.txt. The group sunspot number was downloaded from the NGDC website: https://ngdc.noaa.gov/stp/solar/ssndata.html. The modern F10.7 index was from



http://spidr.ngdc.noaa.gov/. The simulated data by the GCITEM-IGGCAS model under different control runs are available at: http://doi.org/10.17605/OSF.IO/ZQ8HY.

## Competing interests

The contact author has declared that neither of the authors has any competing interests.

## Acknowledgements

The authors acknowledge the support of the B-type Strategic Priority Program of the Chinese Academy of Sciences (Grant XDB41000000), the Project of Stable Support for Youth Team in Basic Research Field, CAS (YSBR-018), the National Natural Science Foundation of China (41621004, 42241106, 42204165), the CAS Youth Interdisciplinary Team (JCTD-2021-

05), and the Key Research Program of the Institute of Geology and Geophysics, CAS (Grant IGGCAS-201904).

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

**Figure 1:** Time evolution of the global mean neutral density and temperature at 400 km in the March equinox (top row). The other rows are global plots of CR4 simulated neutral density (left column) and temperature (right column) for four selected years at UT 19 during the March equinox. The corresponding model drivers are also given in the white text, and 'DM' means

the dipole moment of the geomagnetic field. The gray line in each color plot marks the inclination equator for the corresponding year.



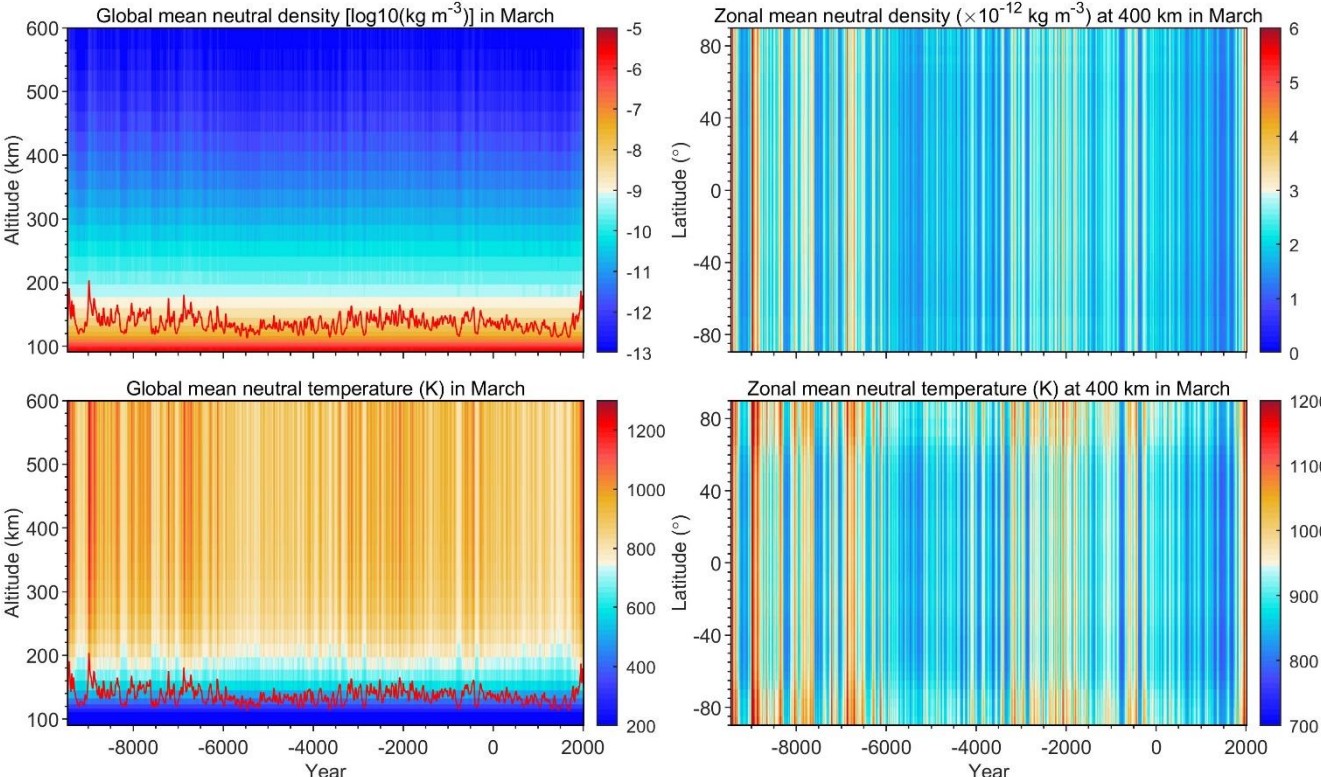

**Figure 2:** Global mean (all UTs and grids) neutral density and temperature profiles versus years (left column) and zonal mean
(all longitudes and UTs) neutral density and temperature at 400 km as a function of latitude and years (right column) in the
March equinox from CR4 simulations. Note that the color scale is different for each plot. The red line in the left column
represents the relative change of the F10.7 index over the Holocene.




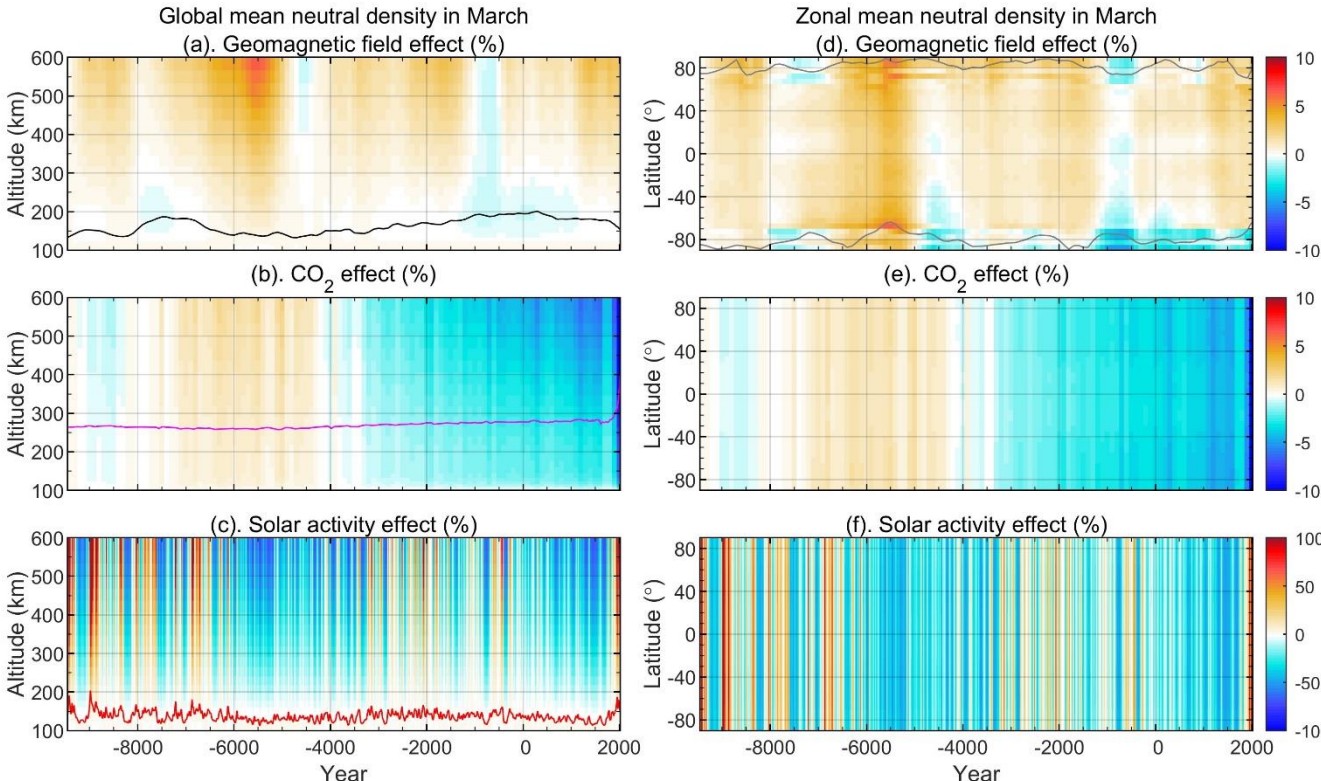

**Figure 3:** The global mean (all grids and UT) neutral density profiles deviate from the beginning year of the simulation due to changes in the geomagnetic field (a, CR1), $CO_2$ (b, CR2), and solar activity (c, CR4–CR3) as a function of altitude and years. Panels (d), (e), and (f) show the same pattern as the left column except that the zonal mean neutral density (all longitudes and UTs) at 400 km is shown. The grey lines in panel (d) mark the latitude of the north and south magnetic poles for the corresponding year. The black, magenta, and red lines in the left column represent the relative changes of the diploe moment, $CO_2$, and F10.7 index, respectively, during the Holocene.



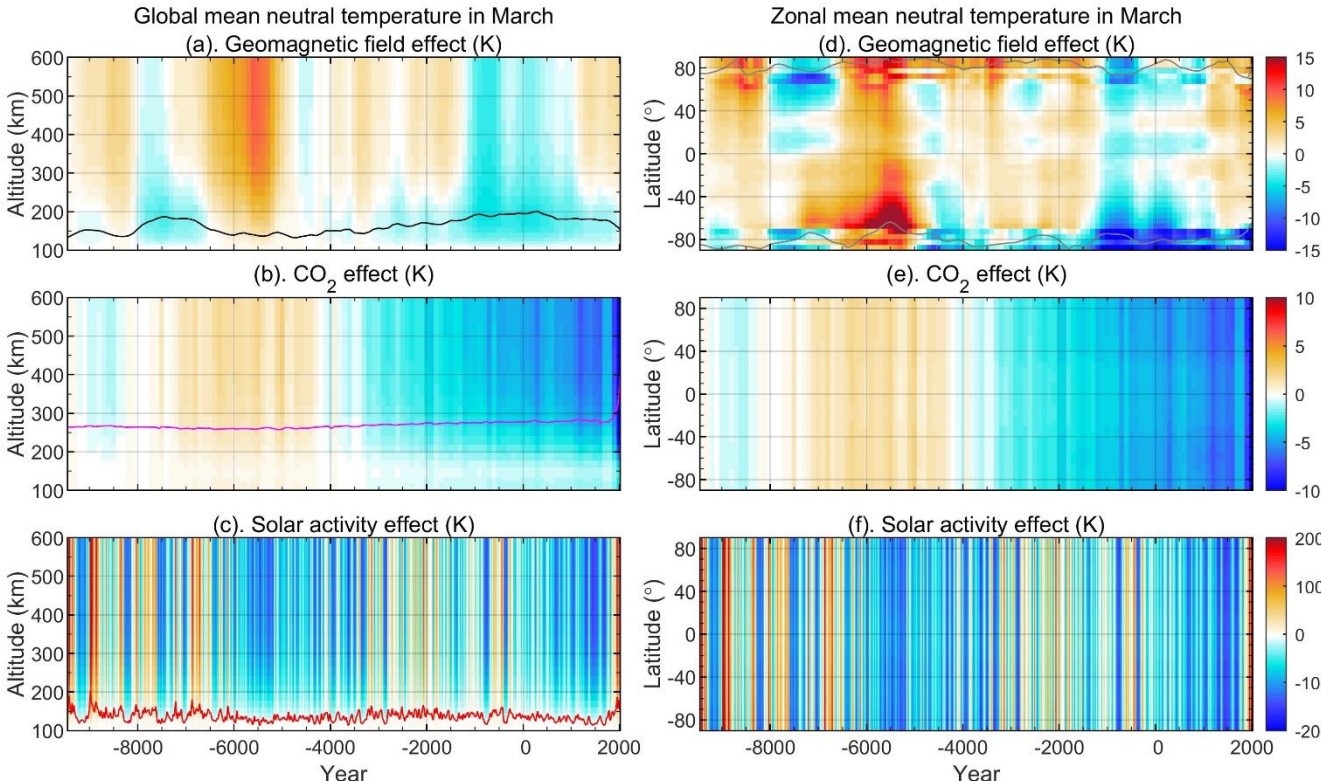

**Figure 4:** The same as Figure 3, but for the neutral temperature.



**Figure 5:** Global mean (red line) neutral density (a) and temperature (b) deviations with respect to the beginning of the simulation versus years at 400 km during the March equinox due to the geomagnetic field variation (CR1), the $CO_2$ variation (CR2), and the solar activity variation (CR4–CR3). The grey dots represent different UT results.



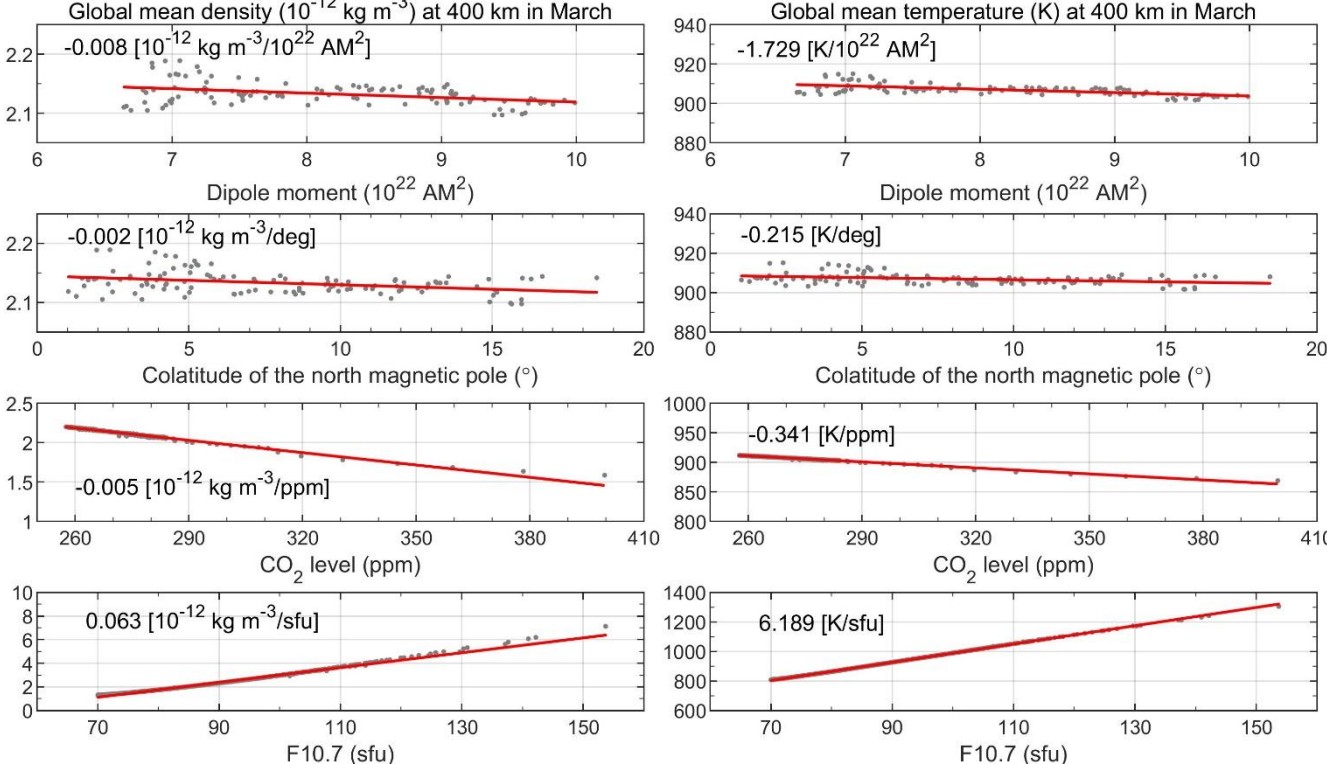

**Figure 6:** Simulated global mean neutral density (left column) and temperature (right column) at 400 km versus the geomagnetic field dipole moment (top row, CR1), the colatitude of the north magnetic pole (second row, CR1), the $CO_2$ level (third row, CR2), and the F10.7 index (bottom row, CR4–CR3) in the March equinox. The red line is the corresponding linear fitting results and the number in each panel is the corresponding fitted linear growth rate.







**Figure 7:** The altitude variations of the linear growth rate (dashed lines) and the corresponding change in percentage (solid lines) of global mean neutral density (left column) and temperature (right column) resulting from the dipole moment (top row, 485 CR1), the $CO_2$ level (middle row, CR2), and the F10.7 index (bottom row, CR4–CR3) in the March equinox (black lines) and June solstice (red lines).







**Figure 8:** The latitude and longitude variations of the linear growth rate of neutral density (b) and temperature (d) due to the geomagnetic field. Panels (a) and (c) are the corresponding linear growth rate of zonal mean neutral density and temperature versus latitude and UT.