# Peer review of "Simulated Long-term Evolution of the Thermosphere during the Holocene: 1. Neutral Density and Temperature"

_EGUsphere, 2023_

## Author Comment (AC1)

**Reply on RC1:**

*This is interesting and useful article, which deserves publication after minor revision. This article reports for the first time simulation of climatic changes in the upper atmosphere over the whole Holocene.*

Thank you for your recognition of our work. We are very grateful for your detailed comments. We have made minor revision to the manuscript accordingly.

**Comments:**

*I would like to see comparison of results of authors with the results of Qian et al. (2021, https://doi.org/10.1029/2020JA029067) over the 1960s – 2010s.*

Qian et al. (2021) indicated that, at 300 km, the change of the greenhouse gas (GHG) concentrations caused a global mean neutral temperature trend with a magnitude of ∼−2 to −3 K/decade, while the change due to the magnetic field driver was overall much smaller in magnitude compared to that due to the GHG driver. The GHG driver caused a global-scale decrease of the mass density at 400 km, on the order of ∼−4%/decade on a global average basis. The change of the magnetic field had a very minor impact on the mass density trend, <10% compared to the impact by the GHG driver. Contributions of the magnetic field driver to the global average trends of the neutral temperature and density were negligible on shorter timescales.

Our results show the global mean neutral temperature trend is −0.341 K/ppm at 400 km, considering an increase in CO2 of ~90 ppm from 1945 to 2015. Therefore, the trend is also ∼−3 K/decade in the last 100 years. For the global mean neutral density, the trend due to the increase of CO2 is about ∼−2%/decade at 400 km. The trends in both global mean neutral density and temperature due to magnetic field changes are negligible.

In addition, Qian et al. (2021) suggested that the magnitude of the neutral temperature trend caused by the change of the GHG concentrations increased with altitude, transitioning to a nearly uniform cooling on the order of ∼−20 K from the 1960s to 2010s, which is generally consistent with our results that the uniform cooling on the order of ∼−26 K from 1945 to 2015, see Figure 1.

[Figure]

Figure 1. The altitude variation of the trend of global mean neutral (a) temperature and (b) density from 1945 to 2015 caused by (black line) CO2 and (red line) magnetic fields.

Overall, our results are generally consistent with the results of Qian et al. (2021), especially in the trend of global mean neutral temperature and the negligible contributions of magnetic field changes on shorter timescales. However, the trend of global mean neutral density in our results is half that of Qian et al. (2021), which may be due to the fact that we consider a much longer time scale than Qian et al. (2021) and that we use the high-latitude convection model of Weimer (1996) rather than Heelis et al. (1982).

***Lines 136 and 137: Both weakening (line 136) and increase (line 137) of dipole moment make increase of neutral density???***

Figure 3 shows that when the dipole moment increases beyond~$3\times10^{22}$ Am$^2$ (in ~7500 BC and from ~1500 BC to ~1000 AD), the density will in turn decrease. A small increase in dipole moment is not sufficient to decrease the density.

**Wording and misprints:**

- Line 47: delete comma – "concentrations," should be "concentration"
- Line 58: delete "in detail" – claim that you understand the evolution over 10,000 years is too strong; you report only the first simulation, which does not include motion of magnetic poles (at present important factor)
- Line 65: "as Yue" should be "as that used by Yue"
- "1990" – 1990 or 1900?
- Line 77: delete "that"
- Line 117: "(Afraimovich et al., 2008)" should be "Afraimovich et al. (2008)"
- Line 121: "clear" – better is "clearly visible"
- Lines 204 and 206: "greater" should be "larger"
- Line 213: "an altitude" should be "altitude"

Thanks for your detailed comments. These comments are very useful for our manuscript. It should be 1900 in line 75. We have corrected other wording and misprints in the corresponding lines.

---

## Author Comment (AC2)

**Reply on RC2:**

*This work (Simulated Long-term Evolution of the Thermosphere during the Holocene: 1. Neutral Density and Temperature) presents new and interesting results regarding the paleo-thermosphere, linked to a previous work of the authors (Simulated Long‑Term Evolution of the Ionosphere During the Holocene). The research on long-term trends, as this one which considers in particular the Earth's magnetic field , the $CO_2$ concentration increase, and solar activity variation as possible sources, are always welcome in the community that studies climate change and trends throughout the atmosphere.*

*I consider that this work can be accepted for publication after minor revisions.*

Thank you for your recognition of our work. Your comments make our articles clearer and easier to understand. We are very grateful for your detailed comments. We have made minor revision to the manuscript accordingly.

**Main Comments:**

**(1)** *The thermosphere variation (temperature and density in this case) which responds to solar activity variation has a timescale of 10 to 11 years, while the variations linked to $CO_2$ increase and/or Earth's magnetic field variation have a timescale of around 100 years. If you consider solar activity variation of the same timescale you have for example, the Gleissberg cycle, whose amplitude is much much weaker than the quasi- decadal variation.*

*I think that this is the change in solar activity which would be interesting to compare with the variations linked to $CO_2$ and the geomagnetic field variations. Since at these time scales I am quite sure they will be all comparable. You have even the Suess-cycle in solar activity to consider also.*

*I consider that the 80% variation in the thermosphere due to the solar activity quasi-decadal cycle is already well known and also that it is a dominant variabillity in the case of inter-annual varaibility.*

***Anyway I consider also important the comparison of all the forcings anaylzed by the authors, even with the solar activity timescale of variation much different than that of the other forcings.***

Thanks for your comment. The characteristic time scales of the Gleissberg cycle and the Success cycle are from 60 to 150 years and from 180 to 250 years, respectively. Therefore, we average our decadal resolution results over 100 and 200 years to obtain the effect of these two cycles on the thermosphere, as shown in the following Figure 1. We can find that the neutral density oscillation is no longer as dramatic as in Figure 3c in our manuscript and is mostly within ±40%, especially in the Suess cycle effect. This is also reflected in the change of neutral temperature in the following Figure 1. However, these features are still mainly controlled by solar activity and are stronger than the effect of $CO_2$ and the geomagnetic field. Therefore, we have not included this information in the revised manuscript. But we still appreciate your comment, which has made us learn a lot.

[Figure]

Figure 1. The effects of (left column) Gleissberg cycle and (right column) Suess cycle on neutral (top row) density and (bottom row) temperature. The red line is the relative change in the 100- or 200-year average of the F10.7 index.

**(2)** *In figures 3 and 4 it is evident that in the last 2000 years the density and temperature variations due to Earth's magnetic field in the magnetic pole regions are opposite. In a pure dipolar field this should not happen, so I guess this is due to the multipolar components of the field (see for example Zossi et al. (2020). Geomagnetic field model indicates shrinking northern auroral oval. Journal of Geophysical Research: Space Physics, 125, e2019JA027434. https://doi.org/10.1029/2019JA027434). This is reasonable since as time passes, and the dipolar component decreases, the Earth's field is less and less dipolar. So the symmetry between northern and southern hemisphere should also decrease.*

*Could the opposite behavior along the last ~2000 years, or so, be related in your case also to the radial component of the field which, due to the multipolar components for example, is increasing in the northern hemisphere and decreasing in the south ?*

*Although, I think that this should lead to lower temperature in the northern pole and higher at the south. Which is opposite to your results.*

*Maybe I am wrong with this reasoning, but if not, I would like the author to comment on this possibility.*

Thanks for your comments. In our other work (Simulated Long-term Evolution of the Thermosphere during the Holocene: 2. Circulation and Solar Tides, https://doi.org/10.5194/egusphere-2023-234), Figure 5 (also Figure 2 below) shows that the non-dipole component of the magnetic field is responsible for the asymmetry of the thermospheric variations between the northern and southern hemispheres. The southern magnetic pole has drifted very little in the last 70 years, so the lower dipole moment leads to an increase in temperature near the southern magnetic pole, which is consistent with your reasoning, but Figures 3 and 4 in our manuscript show the zonal mean for all longitudes and UTs, so it is shown as a decrease in temperature at high latitudes of the southern hemisphere. On the other hand, the long drift distance of the northern magnetic pole leads to a more complex thermospheric variation at high latitudes in the northern hemisphere, which is caused by a combination of magnetic pole drift, neutral wind changes, decreased dipole moment, and increased particle precipitation during the shrinking of the aurora oval.

Overall, your reasoning is very reasonable, and the results of our zonal mean are also reasonable, and we have added some sentences in the revised manuscript to reveal this point, see lines 179–184.

[Figure]

Figure 2. Geographic distribution of neutral temperature (color contours,) and horizontal winds (black arrows) at 350 km in the (a) March and (c) June at UT00. (b) Differences in neutral temperature and horizontal winds are caused by the changes of geomagnetic fields between 1945 and 2015. The scales of wind velocity are labeled in the bottom-left corner of each plot. The changes of magnetic north and south poles between 1945 and 2015 are illustrated in plots (b) and (d).

**(3) *In Figure 8, why are there differences between the two panels ? Is the panel which shows the varaibility with UT for a fixed longitude or a zonal mean ?***

In Figure 8, panels (a) and (c) show the variability of the zonal mean with latitude and UT, and panels (b) and (f) show the variability of the UT mean with latitude and longitude.

**Minor comments:**

**(1) In line 49: "A et al., 2012" is may be "Ridley et al., 2012) ?? But I am not sure. Please check.**

The author is "Ercha A" and the last name is "A". The reference information is "A, E., Ridley, A. J., Zhang, D., and Xiao, Z.: Analyzing the hemispheric asymmetry in the thermospheric density response to geomagnetic storms, J. Geophys. Res.: Space Phys., 117, A08317, https://doi.org/10.1029/2011ja017259, 2012."

**(2) Line 259: " Only the effect of the geomagnetic field is strongly dependent on the universal time and geographical location, and the weakening of the dipole moment leading to an increase in Joule heating in the polar region thus make the thermosphere change more than the effect of CO2. Overall, the higher altitude, the larger the effect of the three drivers on the neutral density and temperature."**

**After the word more, in my opinion it lacks a word. For example "more intense" or "stronger".**

**In the second sentence I would write: " Overall, the higher the altitude, ..."**

Your comments are useful to us and we have made revisions accordingly.